# Recent Advances in Two-Dimensional Quantum Dots and Their Applications

**DOI:** 10.3390/nano11061549

**Published:** 2021-06-11

**Authors:** Konthoujam James Singh, Tanveer Ahmed, Prakalp Gautam, Annada Sankar Sadhu, Der-Hsien Lien, Shih-Chen Chen, Yu-Lun Chueh, Hao-Chung Kuo

**Affiliations:** 1Department of Photonics & Institute of Electro-Optical Engineering, College of Electrical and Computer Engineering, National Yang Ming Chiao Tung University, Hsinchu 30010, Taiwan; jamesk231996@gmail.com (K.J.S.); annadamcut@gmail.com (A.S.S.); 2Department of Electrical Engineering and Computer Science, National Yang Ming Chiao Tung University, Hsinchu 30010, Taiwan; tahmed949@gmail.com (T.A.); dhlien@nctu.edu.tw (D.-H.L.); 3Department of Materials Science and Engineering, National Tsing Hua University, Hsinchu 30013, Taiwan; prakalpgautam@gapp.nthu.edu.tw; 4Semiconductor Research Center, Hon Hai Research Institute, Taipei 11492, Taiwan

**Keywords:** two-dimensional quantum dots, transition metal dichalcogenide, sensors, white light-emitting diodes, photodetectors, phototransistors

## Abstract

Two-dimensional quantum dots have received a lot of attention in recent years due to their fascinating properties and widespread applications in sensors, batteries, white light-emitting diodes, photodetectors, phototransistors, etc. Atomically thin two-dimensional quantum dots derived from graphene, layered transition metal dichalcogenide, and phosphorene have sparked researchers’ interest with their unique optical and electronic properties, such as a tunable energy bandgap, efficient electronic transport, and semiconducting characteristics. In this review, we provide in-depth analysis of the characteristics of two-dimensional quantum dots materials, their synthesis methods, and opportunities and challenges for novel device applications. This analysis will serve as a tipping point for learning about the recent breakthroughs in two-dimensional quantum dots and motivate more scientists and engineers to grasp two-dimensional quantum dots materials by incorporating them into a variety of electrical and optical fields.

## 1. Introduction

The field of material science and technology is evolving rapidly and currently provides a major contribution to research in nanoscale science, motivated by the need for novel materials with fascinating properties. The number of dimensions outside the nanoscale range (100 nm) in which particles are confined determines how a material is categorized. Nanomaterials, in general, are exceptionally tiny, with at least one dimension of 100 nanometers or less. Nanomaterials can have several confinement directions, such as one-dimensional confinement (e.g., surface films), two-dimensional confinement (e.g., strands or fibres), or three-dimensional confinement (e.g., particles). Quantum dots (QDs) are the artificial nanocrystals having confinement in all three dimensions with nanometer size that can transport electrons. By changing their sizes, the optical and electrical properties of quantum dots can be effectively tuned, resulting in the emission of specific wavelengths of light [1,2]. Smaller QDs with a diameter of 2–3 nm emit shorter wavelengths, generating colors such as blue and green, whereas larger QDs with a diameter of 5–6 nm emit longer wavelengths, such as orange or red [3,4]. The emission color and wavelength of QDs corresponding to their sizes are shown in Figure 1a; it can be seen that with increasing size, the emission wavelength increases. QDs are mainly composed of groups II-VI, III-V, and IV-VI materials in which the electron-hole pairs (excitons) are spatially confined in three dimensions (i.e., zero degree of freedom) owing to quantum confinement effect (QCE) [5,6]. The zero-dimensional QDs structures of InP QDs are shown in Figure 1b and InP/ZnSe/ZnS core-shell QDs in Figure 1c [7]. In core-shell QDs, the core and shell are usually made up of form II–VI, IV–VI, and III–V semiconductors in CdS/ZnS, CdSe/ZnS, CdSe/CdS, and InAs/CdSe configurations [8,9,10]. Core-shell QDs can mitigate the issue of low fluorescence quantum yield in originally passivated QDs by utilizing the shell to passivate the surface trap states. In addition, the shell protects against environmental changes and photo-oxidative degradation, and provides a second path for modularity. The size, shape, and composition of both the core and the shell can be precisely regulated, allowing the emission wavelength to be tuned over a broader range of wavelengths than for either individual semiconductor.

Quantum wires are electrically conducting wires with the quantum effects in two dimensions influencing their transport characteristics. Since this phenomenon only happens in the nanometer scale, they are often known as nanowires (NWs). For a specific material, the significance of quantization is inversely proportional to the diameter of the nanowire [11]. It varies depending on the electronic properties of the material, especially the effective mass of the electrons. This means that it will be determined by how conduction electrons interact with the atoms in a particular material. Figure 2a shows CdSSe NWs dispersed on a low-index MgF_2_ substrate illuminated with a 405 nm laser emitting various colors from green to red along the length of the NWs [12]. Nanowires have a number of unique properties that cannot be found in bulk or three-dimensional materials. This is due to the fact that electrons in nanowires are quantum confined laterally and therefore occupy energy levels that are different from the traditional continuum of energy levels or bands found in bulk materials. Figure 2b shows the red illumination of CdSe-CdS quantum rods (QRs) upon excitation in the ultraviolet (UV) region [13]. The efficient overlap of electron and hole wave functions in CdSe-CdS QRs with a spherical seed embedded in an elongated shell contributes to the reduction of nonradiactive decay due to excitons trapped in surface defects, resulting in a high quantum efficiency of emission. Furthermore, the quasi-cylindrical symmetry of the CdS shell and the crystal field effect on the radiative charge recombination within the rods would also result in strongly linearly polarized luminescence along their axis.

In two-dimensional nanomaterials (2D), quantum confinement occurs in one direction exhibiting plate-like shapes and includes graphene, nanofilms, nanolayers, and nanocoatings. However, in bulk or 3D materials, there is no confinement in any dimension. When compared to bulk materials, layered 2D materials have a variety of unique features, including a high aspect ratio (surface-area-to-volume ratio) that is strongly related to the number of layers. From a scientific and technological standpoint, 2D materials such as graphene have a number of benefits over bulk 3D materials because their properties may be dynamically tuned via electrical, chemical, electrochemical, and other methods. Increased relative surface area and quantum effects are two major characteristics that cause the properties of 2D materials to differ dramatically from those of other materials such as 3D materials. These factors can change or improve the optoelectrical and mechanical properties of the materials. When a particle gets smaller, the surface has a higher proportion of atoms than the inside. As a result, when compared to bigger particles, nanoparticles have a substantially higher surface area per unit mass. Even with crystalline solids, as the size of their structural components shrinks, the material’s interface area expands, affecting both mechanical and electrical properties significantly. When matter’s size is reduced to the nanoscale, quantum effects can begin to dominate its properties. These can have an impact on a material’s optical, electrical, and magnetic properties, especially when the particle size gets closer to the nanoscale. Quantum dots and quantum well lasers for optoelectronics are examples of materials that take advantage of these features.

Although Cd-based zero-dimensional QDs exhibit excellent properties, 2D-QDs have recently emerged as a paradigm with new possibilities and features with improved applicability due to their planar confinement [14,15,16]. Two-dimensional QDs are QDs or nanoparticles made out of layered inorganic materials or 2D sheets that have a unique luminescence and chemical capabilities due to their inherent 2D structure. The majority of 2D-QDs retain their 2D shape or lattices from their bulk form, but with improved solution dispersibility and surface functionalization capabilities. Two-dimensional QDs are typically made up of only a few layers or even a single layer of material having lateral dimensions of less than or equal to 10 nm. The bandgap of 2D-QDs can be tuned by optimizing their lateral dimension and number of layers. Two-dimensional QDs are very attractive for various applications such as sensing [17,18], catalysis [19,20,21], batteries [22,23], and biological applications [24] owing to their superficial functionalization and better solution processibility. In essence, 2D-QDs can be considered as QDs originating from the structure of 2D sheets with unique luminosity and chemical properties. There are different types of 2D-QDs, consisting of 2D-QDs with a single component, 2D-QDs with two elements, and 2D-QDs containing multiple components. Under 2D-QDs with a single element category, there exists carbon dots [25], phosphorene dots [26], and quasi 2D-QDs [27,28,29]. Carbon dots (CDs) are zero-dimensional carbon-based materials with significant amount of attention owing to their various advantages such as low toxicity, chemically inert, tunability, good water solubility, and physiochemical properties [30,31]. Only the CDs with 2D structure can be considered as 2D-QDs as they exhibit similar characteristics to that of semiconductor QDs in terms of photophysics and quantum effects. Phosphorene dots are a direct bandgap semiconductor material, primarily black phosphorous with a bandgap lying in between 0.3 and 2 eV and thickness less than 7 nm in its 2D form [32]. Quasi 2D-QDs are those elements of group IV and V such as Si, Ge, B, Pb, and Sn that can be formed in the form of nano-sheets, so they can serve as materials for 2D-QDs. However, there are no experimental studies on 2D-QDs with these materials so far, so there is still an unexplored promise and endless possibilities in the 2D-QDs family domain. Two-dimensional QDs containing two elements consist of Silicon Carbide (SiC) dots, C_3_N_4_ dots, Boron Nitride (BN) dots, transition metal dichalcogenides (TMDs) QDs, and transition metal oxides (TMOs) QDs. SiC dots are wide bandgap semiconductor QDs, which is very attractive for biomedical applications with advanced photoluminescence and long fluorescence lifetime due to efficient quantum confinement [33]. The g-C_3_N_4_ monolayer may be considered to be a graphene sheet containing units of tri-s-triazine connected by the amino group and may serve as another type of 2D-QDs [34]. BN quantum dots are analogous to C_3_N_4_ except carbon (C) is replaced by boron (B), resulting in honeycomb-free structure unlike in g-C_3_N_4_. BN dots show great potential for biological and optoelectronic applications owing to its wide bandgap of 5–6 eV and high quantum yield. TMDs dots are heavy metal-free quantum dots with a general formula of MX_2_, where M can be Mo, In, Pt, Cd, Ti, W, V, Nb, Ni, Re, Rb, Pb, Bi, Ta, Zr, Fe, and Hf; and X can include S, Se, and Te [35,36,37]. MoS_2_ QDs, which display remarkable metallic and other optical properties with the potential to drive significant developments in 2D material-based research, are the most common TMDs QDs. TMOs QDs are typically semiconductor QDs with a larger bandgap than that of TMDs with a general formula of MOx, where M can be Mo, Cr, Sc, W, etc., of which MoO_x_ (x < 3) and WO_3-x_ are the most investigated QDs [25,38,39]. Figure 3a represents the crystal structure of TMDs QDs that is composed of three atomic planes and often two atomic species: a metal and two chalcogens [40]. The honeycomb hexagonal lattice has three fold symmetry and can be used to establish mirror plane and inversion symmetry. At submicron scales, 3D materials no longer behave like their 2D counterparts, which can be advantageous. TMD monolayers are structurally stable, have a specific bandgap, and electron mobilities that are comparable to those of silicon, allowing them to be used in the fabrication of transistors [41]. Figure 3b shows representative scheme of the section of a field effect transistor based on a monolayer of MoS_2_. TMD monolayers are also particularly promising for optoelectronics applications due to their direct bandgap. Atomic layers of MoS_2_ have been used as a photodetector and multilayer MoS_2_ can be used to fabricate an ultrasensitive detector to be with a photoresponsivity reaching 880 AW^−1^, which is 10^6^ times higher than the first graphene photodetectors [42]. Figure 3c shows a typical scheme of the section of ultrasensitive photodetector based on a monolayer of MoS_2_. 

Under 2D-QDs with multiple components, there are MXene-type QDs and other 2D-QDs prepared from MXene-type materials. MXene-type QDs are the two-dimensional transition metal carbides, carbonitrides, and nitrides-based QDs with a general formula of M_n+1_X_n_T_x_ (*n* = 1–3), where M represents the transition metals, X represents the C or N elements, and T_x_ represents the surface termination groups such as –O, –F, or –OH [43]. Other 2D-QDs derived from MXene material include ternary 2D materials such as h-BNC [44], which is a hybridized boron nitride and graphene domains, doped TMDs [45], lamellar metal hydroxides [46], heterostructured TMDs [47], etc. Moreover, two-dimensional (2D) graphene has recently emerged as graphene quantum dots (GQDs) with impressive properties, the bandgap of which can easily be tuned to transform into zero-dimensional quantum dots. By allowing greater surface area, higher solubility, and flexibility of doping with other nanomaterials, the dimensions of GQDs can be modified to provide higher stability characteristics [48,49]. As the family of 2D materials continues to grow and the photoluminescence (PL) of the 2D-QDs is controlled by certain variables such as QCE, surface states, defects, doping, etc., it is essential to study the 2D-QDs PL. QCE is size-dependent and comes to existence in semiconductors when the sizes become comparable to the Bohr radius, resulting in the formation of low-dimensional materials with more advanced features as compared to bulk materials. The PL phenomenon in 2D-QDs can be explained by taking examples of two classes of CDs, one with graphitized carbon core (class 1) and the other with disordered carbon core (class 2). For the class 1 CDs, the energy gap between the highest occupied molecular orbital (HOMO) and the lowest unoccupied molecular orbital (LOMO) decreases with increasing size; however, a contrasting behavior for energy gap dependence on size has been observed for class 2 CDs. This inverse trend is due to the increased strain experienced by the excited state triggered by the disordered carbon core when the QD size is decreased. Hence, as the size of the QDs changes, the trend in the energy bandgap of 2D-QDs would influence the PL. The interaction between the 2D-QDs and the solvents also affects the characteristics of the PL as a strong surface interaction can lead to the luminescence and polarization properties being improved. It has been previously stated that the emission from surface energy traps will lead to the blue PL, while the red PL emission will result from the intrinsic emission from the QCE [50]. In addition, a significant amount of oxidation could generate more surface defects that serve as recombination centers for the excitons, which can influence the PL characteristics, resulting in a narrowing energy gap [51]. Besides oxygen-based groups, carboxyl or amine-based functional groups could cause major surface distortions by acting as non-radiative electron-hole recombination centers, thus affecting the PL characteristics [52]. The fluorescence characteristics of the 2D-QDs also rely on the difference in energy between the intrinsic state and the edge state, creating a significant influence on the 2D material properties [53]. Another important factor affecting the PL characteristics is the doping effect as sulfur or nitrogen doping could lead to a blue-shift, whereas the PL emission peak may experience a red-shift if boron or fluoride is used for doping. Two-dimensional QDs of graphenes and other materials, as well as optoelectronic devices based on them, have attracted a lot of attention in recent years. Many researches have claimed that MoS_2_ 2D-QDs have strong photoresponsivity and detectivity due to outstanding light absorption and a decent PL quantum yield [54,55]. Photodetectors were expected to perform better with a hybrid technique that combined a high light absorption layer with a 2D layer structure. Phototransistors with a hybrid structure of PbS QDs and 2D WeS_2_ have been reported with outstanding performance and broadband photodetection [56]. Another hybrid structure integrating core-shell QDs and WS_2_ nanowalls exhibits great nonradiative energy transfer and excellent No_2_ gas sensing performance [57]. Through plasmonic enhanced photoluminescence, hybrid zero-dimensional core–shell CdSe/ZnS QD/two-dimensional monolayer WSe_2_ semiconductors with an Ag nanodisk (ND) can achieve a high color conversion efficiency of 53% [58]. A detailed explanation of these recent advances is provided in the following sections. 

With all these features in various aspects, 2D-QDs have started to have significant interest in a wide range of applications in recent years, such as sensors, batteries, white light emitting diodes (WLEDs), supercapacitors, photocatalysis, photodetectors, etc. While great advancements have been made, there are some missing remnants that are still struggling to be explored. In addition, in terms of material synthesis, the current growth, efficiency, and productivity of 2D-QDs are far from the criteria necessary for high output and mass production to be met for commercialization in different industries. Figure 4 gives an overview of the organization of the present 2D-QDs review. In this review, we discuss in detail the characteristics, synthesis of 2D-QDs in terms of recent development in various applications, and the challenges faced in terms of device applications.

## 2. Characteristics of 2D-QDs

The properties of 2D-QDs are quite fascinating in terms of broad emissions ranging from the deep UV to near-infrared (NIR) wavelengths that mainly depend on the size-dependent QCE, surface interaction with solvents, defects from ligands, synthesis temperature, etc. Two-dimensional QDs suspensions can emit blue or green light or sometimes red under UV irradiation, but they are transparent and colorless or light yellow and brown under daylight. As we have already mentioned, the size and the surface chemistry are critical factors for determining the PL of 2D-QDs. To mention a few examples, the green light emission from GQDs is attributed to the defect centers originated from ligands such as oxygenous functional groups, while the blue emission is due to intrinsic states in the highly crystalline structure. In addition, solvent types play an important role in deciding the PL color as green emissions will result from methanol, yellow from ethanol, and orange from hexanol. The explanation behind this is due to the fact that the size of QDs varies with the change in solvents and hence the emission wavelengths change. The following section describes the structural and optical characteristics of 2D-QDs in particular.

### 2.1. Excitons in Monolayer TMDs

Excitons in most semiconductors have a small binding energy, usually about 10 meV, and a broad radius, which encompasses several atoms, making them Wannier–Mott type excitons. They are so-called “free” excitons, which are delocalized states that are free to move around the crystal as a single entity. Since the binding energies of excitons in most conventional semiconductors are usually in the same order or lower than thermal energy at room temperature, the excitons are easily dissociated by interactions with phonons. As a result, observing excitonic effects in conventional semiconductors at room temperature is challenging. TMD crystals, on the other hand, have a poor van der Waals layered structure and a large effective mass of valence/conduction bands, resulting in a high exciton binding energy and prominent excitonic effects in transitions at room temperature [59]. The more evident excitonic effects in TMD monolayer is attributed to substantial enhancement of Coulomb interactions in the 2D limit due to spatial confinement and poor dielectric screening [60,61]. This can lead to interesting many-particle phenomena including the creation of various forms of excitons, such as optically allowed and forbidden dark excitons, as well as spatially distributed interlayer excitons states. Figure 1a represents a schematic diagram of exciton, trion, and biexciton generation in TMDs through photoexcitation [62]. The valence and conduction band peaks of the monolayer TMDs are localized at the corners (K and K’ points) of the 2D hexagonal Brillouin zone, resulting in having a direct bandgap. Similar to graphene, this leads to the formation of two inequivalent valleys at the K and K’ points in momentum space. The lack of inversion symmetry causes spin–orbit coupling in monolayers, which induces valence band splitting, and seems to have no effect on conduction bands. As a result, two exciton peaks, A and B, will form owing to the existence of two possible vertical transitions from two spin-orbit split valence bands to a doubly degenerate conduction band. Figure 5b shows the photoluminescence (PL) and differential reflectance spectra of monolayer TMDs flakes on quartz substrate where A and B represent the resonance peaks corresponding to excitonic transitions [62]. For ultrathin samples, differential reflectance is an important indicator of absorbance. Because of the fast intra-valence-band relaxation processes leading to dominant formation of A excitons, the emission of A excitons is much more intense than that of B excitons. Near-degenerate exciton states induce strong absorption at higher energies (C peak). The monolayer TMDs allow the formation of excitons from both same and multi-valleys in the Brillouin zone by coupling between electron and hole in different valleys, as shown in Figure 5c [62]. The former exciton is considered a bright exciton since it normally recombines radiatively, while the latter exciton is called a dark exciton since direct recombination to emit a photon is prohibited as only cross recombination is allowed [63]. Excitons in monolayer TMDs are of Wannier type, which have localized wave functions and can move freely throughout the crystal despite their extremely high binding energy. The energy dispersion curves of exciton, trion, and biexciton have a parabolic shape, as shown in Figure 5d, reflecting the exciton’s freely translational motion [62]. However, because of their heavier masses, trion and biexciton have a smaller curvature than that of exciton. The increase in exciton energy caused by the addition of one more charge carrier characterizes the trion binding energy, E_b,T_ while biexciton binding energy, E_b,xx_ can be defined as the decrease of two isolated excitons after formation of a bound state. Hence, the trion binding energy E_b,T_ can be defined as the subtraction of exciton state energy, E_x_ from trionic state energy, E_T_ and the biexciton binding energy, E_b,T_ can be defined as E_xx_-2E_x_ [64], where E_xx_ is the biexcitonic state energy. In PL or absorption/reflection spectra, the trion (biexciton) binding energy is typically derived by separating exciton resonance energy from trion (biexciton) resonance energy [65,66,67,68]. Unlike exciton binding, which is strongly influenced by the electron-to-hole effective mass ratio, the energy of trion and biexciton binding is hugely affected by the electron-to-hole effective mass ratio.

### 2.2. Structural and Optical Characteristics of Graphene Quantum Dots (GQDs)

Graphene quantum dots (GQDs) have attracted a lot of attention as one of the most important issues in graphene-based electronics due to the QCE at the nanometer scale, which enables researchers to investigate new structural, optical, and electrical phenomena not seen in other materials [69,70,71,72]. Because of its size-controllable characteristics, graphene may generate tunable light in the visible range at the nanoscale level, which is ideal for optoelectronic device applications. The bandgap of GQDs is proportional to 1/L, where L is the average size of GQDs, and can be regulated up to about 3 eV [73]. The electrons in graphene behave like massless Dirac fermions because their valence and conduction bands overlap at two inequivalent Dirac points and exhibit linear spectra near these points. With significant innovations, different sizes and shapes of graphene, such as hexagonal zigzag, hexagonal armchair, triangular zigzag, and triangular armchair, can now be obtained from bulk material [74]. GQDs’ electronic properties are highly influenced by their size and shape as their bandgaps decrease tediously with increase in number of atoms [75]. Since the absorption edges of GQDs can be configured by varying their sizes, they are especially interesting for light harvesting in photovoltaic devices. Figure 6a shows the HRTEM images of GQDs for their major shapes for a given size (d) of GQDs in percentage (p), where p is defined as the ratio of GQDs with a major shape at each average size [76]. Average sizes of GQDs are being shown using parenthesis in Figure 6a. At d = 5 and 10 nm, circular and elliptical GQDs with average sizes of 5 and 12 nm, respectively, are obtained. At each d, circular GQDs account for more than half of the total number of GQDs in these samples. Circular GQDs disappear at d = 15 nm, leaving only elliptical GQDs with one-third of them deformed. Most GQDs are hexagon-shaped at d = 20 nm, but about a quarter of them are distorted with curved sides. The majority of GQDs are hexagon-shaped at d = 25 nm, with small, irregular-shaped GQDs. Most QDs have rounded vertices and are formed like parallelograms with rectangular shape at d = 35 nm. 

The photoluminescence (PL) characteristics of GQDs have been extensively studied because of their potential applications in optoelectronics field. The PL seen in GQDs is caused by electron-hole pair recombination in quantum-confined GQDs, with the peak energy and shape of the PL spectra highly influenced by the size of the GQDs. Figure 6b shows size-dependent PL spectra excited at 325 nm for GQDs of 5–35 nm average sizes in DI water [76]. The sharp PL peak at 365 nm corresponds to PL generated by DI water [77]. The inset of Figure 6b displays various PL colors for different sizes of GQDs. Regardless of the excitation wavelength, the PL spectra of GQDs with an average size of 17 and 22 nm are resolved in two PL bands, presumably due to the integration of GQDs with various sizes/shapes. In addition, as the excitation wavelength is increased from 300 to 470 nm, the PL peak shifts to longer wavelengths consecutively for all GQD sizes. Figure 6c shows the dependence of PL peak shifts on the excitation wavelength from 300 to 470 nm for GQDs having average size in the rage of 5–35 nm [76]. With the exception of 470 nm, all PL spectra display identical size-dependent peak shifts, regardless of excitation wavelength. As the average size increases up to 17 nm, the PL peak energy drops, confirming the QCE. However, the QCE is no longer relevant for average sizes greater than 17 nm, so the PL peak energy increases with increasing size. As long as GQDs maintain their circular/elliptical form for average sizes less than 17 nm, the PL peak energy decreases with increasing size; however, when the shape of GQDs becomes polygons for average sizes greater than 17 nm, the PL peak energy increases with increasing size as shown in Figure 6a. The development of an excited-state relaxation channel resulting in inelastic light scattering or the alleviation of thermalization due to electron–phonon scattering by using an ultrafast high-power light source have been proposed as explanations for the visible PL contained in graphene sheets. In order to investigate the light emission phenomenon in GQDs, the electronic transitions can be controlled by varying the size/shape of the GQDs while producing strong visible PL emissions. Fast carrier-carrier scattering dominates electron–phonon scattering in GQDs, allowing for direct recombination of excited electron-hole pairs, resulting in high-energy PL in GQDs.

## 3. Synthesis of 2D-QDs 

Synthesis of 2D-QDs can be accomplished using one of the two main approaches, top-down and bottom-up approaches. Top-down approaches involve use of physical, chemical electrochemical, and even mechanical methods in order to exfoliate, scale, and decompose bulk materials in well-managed experimental environments for obtaining 2D-QDs. This approach may incorporate intense conditions such as elevated temperatures and chemical reagents such as concentrated acids and strong oxidizing agents. Methods comprising the top-down synthesis approach can sometimes be simpler and cheaper, but it is quite challenging to control the size and shape of 2D-QDs synthesized using these approaches. Bottom-up approaches, on the contrary, implicate synthesis of 2D-QDs with the help of atomic as well as molecular precursors. Bottom-up approaches usually offer better control over size as well as morphology of the 2D-QDs and facilitate efficient utilization of the precursors atoms or molecules. Specific examples on top-down and bottom-up synthesis of 2D-QDs are presented in the following sub-sections. 

### 3.1. Top-down Approaches

Many practical demonstrations of top-down synthesis of 2D-QDs can be found in the literature. Qiao et al. prepared 2D-QDs from monolayer molybdenum disulfide (MoS_2_) using an effective multi-exfoliation method depending on lithium (Li) intercalation [78]. As per the reported demonstration, the monolayer MoS_2_ was made to undergo the first intercalation by dipping of pristine 2H-MoS_2_ powder in n-butyl lithium solution in hexane. Filtration and repeated washing with hexane were performed in order to remove excess lithium as well as organic residues and to retrieve the resultant chemical specie, Li_x_MoS_2_. First, exfoliation was performed immediately after the first inculcation by stirring Li_x_MoS_2_ in water in an ultrasonic stirrer. Adding hydrochloric acid reduced the pH value of the solution to 2, which made flocculation occur rapidly. The resultant mixture was washed with the water and centrifuged a few times to attain a solution with a pH value of 7 corresponding to the neutral flocculation. The dried, re-stacked MoS_2_ nanosheets were then made to undergo the second and third exfoliation with Li-intercalation by reiterating the same procedure as mentioned above. Un-exfoliated material was removed by centrifuging the resulting mixture and purification of the mixture was accomplished by removing lithium hydroxide (LiOH) with the help of a dialysis bag. Finally, MoS_2_ QDs were collected by further centrifuging the mixture and then annealed to obtain enhanced photoluminescence. The complete process to extract 2D-QDs from MoS_2_ is summarized in Figure 7.

Another top-down approach of synthesizing 2D-QDs is reported by Gopalakrishnan et al. who demonstrated synthesis of MoS_2_ quantum dots interspersed in few-layered sheets of MoS_2_ [79]. Heterodimensional MoS_2_ QDs were prepared by incorporating a liquid exfoliation technique in various organic solvents. To briefly describe the experimental approach, MoS_2_ powder was mixed with 1-methyl-2-pyrrolidone in a container and sonicated with the help of an ultrasonic bath uninterruptedly for a few hours. Then, the resulting dispersion was sonicated with a sonic tip for a few more hours. The dispersion was centrifuged again after keeping undisturbed for a whole night. Temperature was maintained below 277 K by placing the sample in an ice bath. A step-by-step description of the procedure is depicted schematically in Figure 8.

### 3.2. Bottom-up Approaches

Top-down synthesis approaches, discussed in the previous sub-section, suffer from a few major limitations. These approaches are usually affected by environmental conditions, require expensive and hazardous organic solvents, and ask for intense pretreatment measures. Moreover, it is challenging to precisely control the characteristics and morphology of the 2D-QDs produced using top-down approaches. An alternative method to synthesize 2D-QDs, a so-called bottom-up approach, makes use of atomic or molecular precursors to form the 2D-QDs with desired characteristics. Ideally, bottom-up growth of 2D-QDs (or other low-dimensional nanomaterials) can be prepared with almost infinite number of possible precursors but the challenge in this regard is to develop sufficient understanding of the reaction energetics and precisely control the reaction dynamics [80]. Figure 9a shows schematically how a nanosheet can be grown from molecular precursors, incorporating a bottom-up synthesis approach [81]. Wang et al. report on preparation of MoS_2_ QDs under hydrothermal conditions by incorporating sodium molybdate and cysteine as precursors [82]. The steps to accomplish the synthesis are summarized as follows: Na_2_MoO_4_·2H_2_O was dissolved in water, pH of the solution was attuned to pH 6.5 with the help of 0.1 M HCl after ultra-sonication, water and L-cysteine were added to the solution followed by a few minutes of sonication, the resulting mixture was then transferred into Teflon-lined stainless steel autoclave and made to react while maintaining the temperature at 200 °C, the solution was cooled in ambient environment, and supernatant containing MoS_2_ QDs was gathered after a few minutes of centrifugation. The process preparing 2D MoS_2_ QDs and their use as a sensor for trinitrophenol (TNP) detection is depicted schematically in Figure 9b [82].

In summary, 2D-QDs can be prepared using both top-down and bottom-up approaches while being mindful of the pros and cons of each individual approach. Top-down approaches usually offer the benefit of inexpensive exfoliation targets while bottom-up approaches have potential to create 2D-QDs with well-controlled characteristics. Top-down approaches require intense reaction conditions and strong chemical reagents while bottom-up methods are relatively less mature and more sensitive as well as challenging.

## 4. Hybrid 2D Quantum Dots Materials and Their Applications

Two-dimensional QDs such TMDs, GDQs, and others have recently sparked a lot of interest due to their peculiar optical and electronic properties such as tunable energy gaps, effective electronic transport, and semiconducting characteristics [37,83,84,85]. These 2D-QDs have received considerable attention since the advent of graphene and tunable size/shape GQDs and have become a cutting-edge research subject. Owing to their ease of functionalization, 2D-QDs such as TMDs have piqued researchers’ interest for a variety of applications, including transistors, optoelectronic devices, catalysis, energy conversion, batteries, and sensing [41,86,87,88,89,90,91,92,93,94,95,96,97]. Conversely, atomically thin TMDs have low light absorption, so researchers are working on possible strategies to increase the performance of photodetectors based on TMDs. Subsequently, many researchers proposed various nanostructures such as nanobelts, nanorods, and vertically aligned layers to improve the aspect ratio and light-harvesting abilities of TMDs [98,99,100,101,102]. Another important technique for improving photodetection efficiency is to use heterostructure design or nanoparticle decorations [103,104]. Hybridization of colloidal QDs with TMDs has also been investigated by many researchers to improve optical sensing properties [105,106,107,108]. More specifically, if the emission of donor (QDs) and acceptor (TMDs) have spectral overlap during hybridization, a strong dipole–dipole coupling mechanism, NRET (non-radiative energy transfer), will be triggered between them [109,110]. The magnitude of spectral overlap and the separation between donor and acceptor are the two most important factors for determining NRET efficiency. This NRET efficiency implies an increase in energy transfer rate, which explains why QDs/TMD hybrid devices have better photoelectrical performance [111,112].

In this regard, Tang et al. proposed a hybrid 3D nanostructure for optical sensing and NO_2_ gas-sensing by integrating colloidal CdS/ZnS or CdSe/ZnS core-shell QDs with 3D WS_2_ nanowalls [57]. This method transforms a WO_x_ thin film into vertically standing WS_2_ nanowalls using a quick synthesis method with high heating and cooling rates. Figure 10a depicts an illustration of the WS_2_ nanowalls synthesis process using the chemical vapor reduction method in a horizontal quartz tube furnace. The sulfurization process started with the WO_3_/SiO_2_/Si substrate and sulfur powder as the precursors. After rapid heating to 65 °C and rapid cooling at a rate of approximately 25 °C/min, a high crystal quality WS_2_ film with 3D nanowall structures was developed, which is confirmed by a series of material characterization. The hybrid structures of QDs and TMDs have already been discussed as a way to improve their individual performance, and it is believed that the QDs/TMDs hybrid device will be a good candidate for various optoelectronic applications. Tang et al. demonstrated an innovative design of a core-shell QDs/WS_2_ hybrid device for gas sensing applications. The device is assembled in a gas delivery chamber after being bonded to the electrodes on a printed board circuit (PCB) with silver wires to investigate the sensing application. To facilitate effective NO_2_ gas desorption, a 365 nm UV LED light was installed in the chamber and was continuously switched on and irradiated on the device during measurement. They used a graphene CdSe QDs/WS_2_ hybrid device in this study because it has the best non-radiative energy transfer efficiency. Figure 10b demonstrates the structural model and p–n junction model of CdSe-ZnS QDs on 3D WS_2_ nanowalls. During the gas sensing mechanism, the electrons from n-type CdSe-ZnS QDs near the surface tend to diffuse into p-type WS_2_, while holes from WS_2_ near the interface diffuse into the surface of CdSe-ZnS QDs. As a result, a space charge carrier diffusion region is formed at the p–n interface as a potential barrier, as shown in the second panel of Figure 10b. When the hybrid device is exposed to NO_2_, the NO_2_ molecules interact with the 3D WS_2_ nanowalls, causing electrons to be extracted from the surface. This will cause the equilibrium in the space charge region to be disrupted, causing electrons from the n-type CdSe-ZnS QDs side to diffuse through the space charge region before a new equilibrium is established [113,114]. As a consequence, the potential barrier’s width would be reduced, increasing the conductivity of the hybrid device and resulting in improved sensing efficiency. For monitoring the gas-sensing properties, the gas delivery chamber was pumped to vacuum and the pumping operation was suspended when NO_2_ gas was injected into the chamber during the measurements. Figure 10c represents the time-resolved response measurement of NO_2_ for different concentrations showing the corresponding dynamic NO_2_-sensing curves. As shown in Figure 10d, the hybrid device demonstrates substantial responses ranging from 40.4% to 95.7% at varying concentrations ranging from 50 parts per billion (ppb) to 1 part per million (ppm), respectively. At an exceptionally low concentration of 50 ppb NO_2_, the G-CdSe QDs/WS_2_ device performed admirably. Figure 10e shows the device’s sensing test in a concentration of 50 ppb NO_2_ gas, which confirmed the device’s strong cycling stability at such a low gas concentration. Figure 10f shows the response and recovery times of the G-CdSe QDs/WS_2_ gas sensor at a concentration of 1 ppm NO_2_, suggesting a remarkably quick response time of 26.8 s to achieve the outstanding gas response efficiency of 95.7% and just 187.9 s to return to the restored state. The G-CdSe QDs/WS_2_ device’s overall gas sensing characteristics are equivalent or even better than those recorded in previous studies for TMD-based and heterostructured gas sensors. In comparison to previously described QDs/TMD thin film device configurations, this hybrid system facilitates the integration of a high-aspect-ratio 3D nanowall structure with the characteristics of nanosized QDs, resulting in a much larger contact area between QDs and TMDs, which can also expedite detecting behaviors. Furthermore, the scalability of this hybrid structure enables the indicated device to have a much wider range of applications in advanced sensing.

In addition to sensing applications, ultrathin monolayer TMDs have significant potential for use in the development of flexible light-emitting devices of mini-scale [115,116,117]. The strong spatial confinement and weak dielectric screening in monolayer TMDs would lead to strong Coulomb interaction and exciton formation with exciton binding energy of about 0.32–0.72 eV [65,118,119]. This high binding energy is responsible for making the excitons effects stable at room temperature, which is favorable for delivering strong light–matter coupling. Plasmonic materials have already been shown to improve luminescence in a number of applications by enabling intense light–matter interactions [120,121]. Plasmonic nanostructures are known to enable high electric field confinement, so they can easily be coupled with 2D TMDs to enhance light absorption and emission. When plasmonic nanostructures and TMDC materials are coupled, a plasmon–exciton coupling effect occurs, resulting in a plexciton that can be used to boost photoluminescence (PL) [122,123,124]. In addition, the plasmonic nanostructures can strengthen the absorption mechanisms of TMD materials as the plasmonic resonance spectrum matches the absorption spectrum of TMDs [125,126]. Chang et al. studied the plasmonic-enhanced PL and color conversion efficiency of a plasmonic nanostructure containing a hybrid 0D core-shell CdSe/ZnS QD/2D monolayer tungsten diselenide (WSe_2_) semiconductor with an Ag nanodisk (ND) [58]. Plasmonic nanostructure of an Ag ND array was used to enhance the emission from the monolayer WSe_2_ and improve color conversion from QDs to WSe_2_. Figure 11a shows the schematic of the QD-Ag-WSe_2_ hybrid structure with Ag ND patterned on a monolayer WSe_2_ film that had previously been pre-transferred, and then CdSe QDs were sprayed on top of it. A 5 nm thick alumina (Al_2_O_3_) spacer layer was deposited on the interlayers between QD and WSe_2_ to prevent from PL quenching of the monolayer WSe_2_ due to charge transfer between WSe_2_ and Ag ND. The PL intensity of the QD-Ag plasmonic nanostructure was found to be significantly higher than that of CdSe QDs without the Ag ND. The emergence of exciton–plasmon interactions to improve exciton absorption is credited with this PL boost. Furthermore, since the local surface plasmonic resonance (LSPR) spectrum overlapped with the exciton absorption and emission peak of monolayer WSe_2_, the PL spectrum of Ag-WSe_2_ hybrid structure was found to be enhanced by an enhancement factor of 1.8. To investigate the light color conversion mechanism from the emission of CdSe QDs to monolayer WSe_2_, the CdSe QDs and monolayer WSe_2_ with Ag ND were integrated on a SiO_2_ substrate. Figure 11b illustrates the PL spectra of the three nanostructure samples with an Ag ND array with a diameter of 123 nm. The orange, red, and green solid lines correspond to the QD-Ag, WSe_2_-Ag, and QD-Ag-WSe_2_ nanostructures, respectively. The QD-induced emission from the QD-Ag-WSe_2_ nanostructure had a considerably lower PL intensity than the QD-Ag nanostructure. The PL intensity of the WSe_2_ emission in the AgWSe_2_ nanostructure, on the other hand, was increased after QDs were added. The energy conversion from the QD emission to the WSe_2_ emission was most likely the cause of this enhancement. The color conversion efficiency (η) from QDs emission to WSe_2_ emission was estimated for various LSPR wavelength of the Ag ND plasmonic nanostructures as shown in Figure 11c. The coupling of the Ag ND plasmonic nanostructure and the fine-tuning of the Ag nanostructure plasmonic resonant wavelength resulted in a 53% color conversion efficiency from QDs to WSe_2_. The efficiency was found to decrease with increase in resonant wavelength reaching <1% corresponding to an absorption wavelength of 800 nm. The energy transfer efficiency of bare QD, QD-Ag, and QD-Ag-WSe_2_ nanostructures was investigated using time-resolved PL with varying Ag ND absorption wavelengths from 500 to 850 nm. Figure 11d illustrates the PL decay curves for the QD peak with Ag ND in the QD, QD-Ag, and QD-Ag-WeS_2_ structures fitted by a tri-exponential decay at a resonant wavelength of 650 nm. The QDs in the QD-Ag-WSe_2_ structure have a faster PL decay rate than those in the QD-Ag structure. This faster decay rate of the QDs is attributed to exciton–plasmon coupling effect and PCRET mechanism in the QD-Ag-WSe_2_ structure. The PL decay rates for QDs in the QD, QD-Ag, and QD-Ag-WSe_2_ structures were estimated from the TRPL analysis and found to be 0.99 ± 0.03, 1.78 ± 0.05, and 2.38 ± 0.09 ns^−1^, respectively. The energy transfer rates of the QDs in the QD-Ag-WeS_2_ structure were obtained using the PL decay rates values and plotted in Figure 11e as a function of the plasmon resonance wavelength of Ag NDs. The energy transfer rate of QDs is found to be dependent on the plasmon resonance wavelength of Ag NDs. Furthermore, the energy transfer rate curves exhibit a similar behavior to the color conversion efficiency curve from QD to WSe_2_ emission. The energy transfer rate reaches a maximum of 0.16 ns^−1^, which corresponds to a plasmon resonance wavelength of 650 nm. Subsequently, hybrid QD/TMD light emitters can be coupled with GaN-based white LEDs to improve color temperature and broaden the color gamut of a miniature white LED. 

It has previously been stated that combining QDs and TMDs would result in improved photoelectrical performance due to the coupling mechanism and high energy transfer rate. Two-dimensional TMDs have been opted for photodetector or phototransistor due to its suitable bandgap and high carrier transport mobility [42,127]. As a result, it is anticipated that combining QDs and 2D materials would dramatically improve photodetectors’ performance. Hybrid QDs/graphene phototransistors have been demonstrated to have high responsivity owing to the photogating effect caused by capacitive coupling [128]. Most graphene-based devices, on the other hand, have a high dark current and a slow response time due to the presence of semimetal channels [129,130,131,132]. To address these issues, a 2D MoS_2_ channel has been used to develop a QDs/MoS_2_ hybrid phototransistor, which suppressed dark current and improved response time [133]. Because of its proper bandgap values (direct bandgap of 1.6 eV in single layer form and indirect bandgap of 1.3 eV in bulk form), large surface-to-volume ratio, and high mobility, WSe_2_ is ideal for optoelectronic applications. Hu et al. designed hybrid phototransistors for high performance and broadband detection using high transport mobility p-type 2D WSe_2_ and PbS QDs [56]. Figure 12a depicts the device structure of the PbS QD-capped hybrid back-gate WSe_2_ phototransistor. The transistor channel, which has a width of about 10 m, is made of monolayer WSe2 grown by chemical vapor deposition. A thermal evaporation method with a shadow mask has been used to deposit gold (Au) electrodes (source and drain) of about 100 nm. Colloidal PbS QDs were spin-coated on a WSe_2_ nanosheet device to create the hybrid device, and the coated QDs ligands had to be replaced later with tetrabutylammonium iodide (TBAI). The photocarrier transport mechanism in the hybrid photodetector under infrared (IR) illumination is shown in Figure 12b. Under illumination, photoexcited carriers are generated at the PbS QDs layer and the built-in potential between the PbS QDs and WeS_2_ monolayer separates the carriers at the p–n junction. The photogenerated holes will be moved from PbS QDs to WSe_2_ monolayer under an applied electric field, V_DS_, while electrons will remain accumulated in the PbS QDs layer. These electrons can efficiently gate the WSe_2_ nanosheet by functioning as a photogating component through capacitive coupling and help in regulating device conduction. The hybrid phototransistor’s system output has been investigated, and it has been discovered that it inherited the wide spectral detection due to the broad absorption of PbS QDs. Figure 12c represents the wavelength dependence response curve of the hybrid phototransistor with under-applied voltage of 1 V at back-gate voltage (V_GS_ = 0 V). The response peak at 970 nm corresponds to the first exciton peak. By varying the size of PbS QDs, it is possible to achieve wavelength selectivity and a wide photoresponse for UV–vis–NIR detection. The specific detectivity (D*) of the hybrid phototransistor has also been investigated and plotted in Figure 12d as a function of back-gate voltage under 970 nm illumination. At all back-gate voltage ranges, the specific detectivity could reach 10^13^ Jones, regardless of whether the back-gate was in the ON or OFF state. Figure 12e shows the responsivity and specific detectivity curves as a function of V_DS_ at V_GS_ = 0 V. The high responsivity of the hybrid photodetector is attributed to the superior detectivity properties. The responsivity as well as the specific detectivity increase with increasing applied field at V_GS_ = 0 V with an estimated maximum value of about 7 × 10^5^ AW^−1^ and 7 × 10^13^ Jones, respectively, which is superior to other TMDs-based photodetectors. Future optoelectronic devices may benefit from this current device construction strategy, improved photogating performance, and stable device operating conditions.

Aside from TMDs, graphene QDs has shown to have unique properties for ultrafast photodetectors due to its high charge carrier mobility, thin profile, tunable optical properties, wavelength-dependent absorption, and strong absorption spectrum [134,135,136]. The ultrafast conversion of photons to currents in ultrafast photodetectors is specifically driven by high carrier mobility. Previous research has shown that the most sensitive graphene-based photodetectors require the inclusion of an electrically passive sensitizing layer made up of colloidal QDs (CQDs), perovskites, carbon nanotubes, or 2D materials [128,129,137,138]. Since graphene has zero bandgap, it can generate charge carriers by light absorption across a broad energy spectrum. Furthermore, graphene enables wafer-scale integration by low-cost manufacturing based on complementary metal-oxide-semiconductor (CMOS) fabrication processes. The graphene photodetectors have the advantage of being compatible with standard photonic integrated circuits in terms of the manufacturing process. Unfortunately, graphene photodetectors have low sensitivity and broad dark currents due to biasing of the graphene channel or fabrication method [139,140,141,142,143,144]. As a result, developing a better photodetector configuration that can fully exploit graphene’s properties is a top priority. In this regard, Nikitskiy et al. addressed the challenges by demonstrating a hybrid photodetector device by integrating a CQD photodiode atop a high-gain graphene phototransistor [145]. The carrier drift in the photodiode due to electric field can dramatically increase charge collection by altering the electrically passive sensitizing layer to an active one. Figure 13a illustrates the optical image of the hybrid graphene transistor–CQD photodiode detector, as well as the electrodes used. The device is made up of a graphene channel on top of which is deposited a patterned 300 nm thick CQD film, which is then overcoated with an ITO electrode’s top-contact. In the region where the three layers overlap, the photodetector’s active area is created. The CQD layer is patterned to prevent charge carriers from escaping to the drain contacts, thus obviating the need for graphene. This four-terminal sensing device is being designed to eliminate the contact resistance and isolate measurement only over the photodetector’s active region. Figure 13b shows the phototransistor operation as well as the illustration of the electronic circuit. When light is incident on the device, electron-hole pairs are generated in the CQD layer drifted by the built-in electric field at the interface and by the bias applied across the diode. Electrons will continue to remain in the CQD, while holes will diffuse to the graphene channel, changing its resistance due to depletion at the interface. A voltage drop is estimated between V+ and V− electrodes, while a constant current is driven between source and drain electrodes. Without accounting for the resistance of the metal and graphene contacts, the photo-induced change in resistance or voltage drop can be measured with the highest accuracy. The responsivity and EQE of the indicated device were measured and plotted in Figure 13c as a function of applied top-contact bias (V_TD_). The responsivity of the phototransistor is measured to be 5 × 10^8^ VW^−1^ at V_TD_ of 1.2 V corresponding to bias current of 100 µA. The EQE is found to be dependent on V_TD_ as shown in Figure 13c, where it increases and saturates after certain V_TD_ value. Without any bias, EQE is about 10%, and it increases with V_TD_ to about 75% at 1.2 V reverse bias, where it saturates. This is due to the fact that as V_TD_ increases, the depletion region expands, improving charge collection efficiency. The carrier lifetime of the hybrid photodetector has been estimated from the photoresponse bandwidth shown in Figure 13d. The electrical 3dB bandwidth of the photodetector can reach a maximum value of 1.5 kHz and was found to be increasing with increasing V_TD_, as shown in the inset of Figure 13d. The extracted effective lifetime of the photodetector is 106 µs corresponding to 1.5 kHz bandwidth. In addition, the photoconductive gain has been estimated using decay components and found to be on the order of 10^5^, with a gain-bandwidth product of >1.5 × 10^8^ Hz. Figure 13e illustrates the photoresponse of the photodetector expressed in terms of ΔV as a function of incident irradiance. The lowest observable irradiance was estimated to be 10^−5^ Wm^−2^ at a V_TD_ of 1.2 V, with the detector’s linear dynamic range increasing as V_TD_ values increased. The inset of Figure 13e demonstrates the linearity of photoresponse for high irradiance values. The photoresponse is observed to increase as the incident irradiance value increases, being saturated at a power density of 0.5 Wm^−2^. Figure 13f shows the measured responsivity of the detector in terms of VW^−1^ and AW^−1^ as a function of incident irradiance. The high linear dynamic range achieved by using a photodiode rather than a passive sensitizer is demonstrated by the flat responsivity responses over a broad range of power density. Using a photodiode instead of a passive sensitizing layer allows for a much wider range of output parameters. Finally, this hybrid design demonstrates graphene’s and other two-dimensional materials’ ability to be effectively combined with other optoelectronic materials, paving the way for hybrid 2D/0D optoelectronics.

Min et al. reported another hybrid structure comprising zero-dimensional (0D) GQDs and semiconducting 2D MoS_2_ that exhibits remarkable properties for optoelectronic devices, outperforming MoS_2_ photodetectors [146]. GQDs exhibit unique optoelectronic features such as long carrier lifetimes and rapid electron extraction due to enormous transition energies and weak coupling to excitonic states. When GQDs interact with 2D materials, quantum effects can influence charge carrier dynamics, allowing for charge transfer, carrier separation, and collection. The hybrid GQD/MoS_2_ photodetectors were made by drop-casting the GQDs solution over bulk mechanically exfoliated MoS_2_ membranes on SiO_2_/Si substrates, as shown in Figure 14a. The insets of Figure 14a depict the corresponding molecular structures of GQDs (top view) and MoS_2_ (side view). The photoelectrical mechanism of this device consists of different photo-physical steps—photoexcitation, re-absorption, tunneling, and thermal excitation. When the energy of the incoming photon source exceeds the bandgap of GQDs, photoexcitation occurs in MoS_2_ and GQDs. A re-absorption process of emitted photons from GQDs by MoS_2_ is then detected, thereby increasing the photocurrent by generating more electron-hole pairs. Following that, photoexcited electrons in the conduction band of GQDs are injected into MoS_2_ to initiate a tunneling process. Similarly, holes from MoS_2_’s valence band will be transferred to GQDs, resulting in a higher rate of recombination. Finally, the Schottky barrier formation at the interface of GDQs and MoS_2_ leads to thermal excitation of higher-energy electrons from GQDs to MoS_2_. As a result of the multiple charge carrier amplification processes, the photoresponse of hybrid GQD/MoS_2_ devices will be higher than bare MoS_2_ devices. Using a tunable laser source for optical illumination, the photoresponse of the hybrid GQD/MoS_2_ and bare MoS_2_ devices as a function of wavelength has been analyzed, as shown in Figure 14b. The photoresponsivity of the hybrid GQDs/MoS_2_ was found to be 775 AW^–1^ at a laser wavelength of 400 nm, whereas the photoresponsivity of bare MoS_2_ was found to be 44.8 AW^–1^. In comparison to previous photodetector studies based on other material systems such as CuPc and CdTe, the experimentally determined photoresponsivity is over 300 times higher [147,148]. Furthermore, the GQD/MoS_2_ hybrid device exhibits a detectivity of 2.33 × 10^12^ Jones and an EQE (~241%) of almost 17 times higher than that of the bare MoS_2_ device (~14%).

Two-dimensional TMDs with substantial spin–valley coupling, which introduce a significant coupling between the spin and valley degrees of freedom, have recently attracted a lot of attention for prospective applications in spintronics, valleytronics, and other fields. Furthermore, earlier research has revealed that the lack of interlayer interaction causes an indirect-to-direct bandgap transition in monolayered WS_2_ sheets [149,150]. Because of the strong spin–valley coupling and the presence of a direct bandgap in monolayered WS_2_ sheets, they can be used in optoelectronics, spintronics, valleytronics, and other fields [151,152,153]. Due to the strong quantum confinement effect, monolayered WS_2_ QDs can display more remarkable features than monolayered sheets, which strengthens the spin–valley coupling and widens the bandgap. Zhang et al. demonstrated ultrasmall and monolayered WS_2_ QDs with giant spin–valley coupling and purple luminescence [154]. Three absorption peaks are observed on monolayered WS_2_ sheets: 625 nm, 550 nm, and 450 nm. The absorption peaks at 625 and 550 nm are due to transition from the spin-splitting valence band to the conduction band at the K point of the Brillouin zone while the optical transition between the density-of-state peaks in the valence and conduction bands causes the absorption peak at 450 nm. However, for annealed WS_2_ QDs, three exciton absorption peaks at A (379 nm), B (303 nm), and C (269 nm) are observed. The large energy differential between the A and B excitonic absorption peaks in annealed WS_2_ QDs is discovered to be up to 821 meV, implying the presence of a huge spin–valley coupling. The annealed WS_2_ QDs have two unique emission peaks at 416 and 342 nm, corresponding to A and B emission peaks. It is obvious that the significant quantum confinement effect will undoubtedly enhance the spin–valley coupling and increase the bandgap of QDs. As a result, the ultrasmall lateral size could be responsible for both the enormous energy difference between the two absorption peaks and the purple PL in annealed WS_2_ QDs. Moreover, there is no spin–orbit splitting in WS_2_ QDs and this suppression in the intervalley scattering is also responsible for the giant spin–valley coupling. 

Table 1 summarizes the performance of 2D-QDs-based devices used in this review.

There are numerous pros and cons to employing 2D-QDs in several electrical and optical systems. Two-dimensional materials have been extensively explored as channel materials for future electronic device applications due to their atomically thin structure and superior electrostatic control, which enhances immunity to short channel effects and the loss of band-edge sharpness. With the number of stacked layers and the amount of strain, the bandgap and mobility of these materials changes. On the one hand, being able to tune channel attributes makes many circuit design challenges easier to solve. However, uncontrolled variation might lead to catastrophic yield consequences. Two-dimensional materials are excellent choices for optoelectronic devices owing to their unique features, such as a wide response spectrum, exceptional flexibility, and strong light–matter interaction. Additionally, 2D heterostructure-based optoelectronic memory device capable of accumulating and releasing photo-generated carriers under the influence of an electric field and light irradiation can be developed. Because of the emergence of 2D materials, miniature, flexible, and low-energy optoelectronic storage systems are now possible. Due to their strong charge mobility and moderate bandgaps, field-effect transistors (FETs) have been made from a range of semiconducting 2D materials such as TMDs and black phosphorus, making them suitable candidates for this application. The intrinsic flexibility of 2D materials is an advantage, as they may be used to produce flexible circuits when combined with suitable substrates. Many TMDCs (such as MoS_2_, MoSe_2_, WS_2_, and WSe_2_) and black phosphorus have a bandgap in the optical or near-infrared range and good charge transport properties, making them ideal for high-efficiency photodetectors. Two-dimensional materials have exceptional spin-valley characteristics, allowing for spin injection, manipulation, and detection in a single integrated device, resulting in scalable and ultrafast nonvolatile logic circuits with extremely low energy dissipation. TMDs with a direct bandgap allow circularly polarized light to excite carriers preferentially within a given valley with a specified valley pseudospin. Unfortunately, there are a number of challenges that limit the use of single 2D materials. When metal electrodes are directly deposited on 2D semiconductors during device manufacturing, a Schottky barrier comes into existence, resulting in high contact resistance. Furthermore, due to their short lifetime, intralayer excitons formed in single 2D semiconductor materials are challenging to handle, limiting their use in exciton devices. Most insulated 2D materials, including hBN, are not acceptable for use in devices on their own, and BP is readily oxidized when exposed to air. In the following section, we go over these problems in greater detail.

## 5. Challenges Faced by 2D-QDs Materials 

Despite recent accomplishment in this fascinating research area, there are several crucial challenges to be taken into consideration.

### 5.1. Synthesis/Deposition of 2D-QDs Materials

The first challenge concerns the synthesis of crystalline 2D materials, including the control of their crystalline phases, grain size, grain boundaries, and morphology. The structure characteristics play a very prominent role in the chemical and physical properties of 2D materials. Therefore, a sagacious structure of interface layer synthesis is vital. Consequently, the structure of 2D materials extremely depends on the physical and chemical properties of the precursors, such as solubility, thermal stability, sensitivity, catalytic activity. For a better understanding of the crystal growth mechanism, the structure characterization at the atomic level is a critical challenge. Several in situ characterization techniques such as in situ XRD and TEM have been developed in the past few years. However, expeditiously observing the growth of 2D materials using these techniques is not quite facile. For exemplification, the formation of 2D materials at the interfaces due to the assemblage of the precursor is still inconclusive by TEM caused by sensitivity to electron irradiation [155]. Moreover, the adequately low mechanical and chemical stability causes the defect during the transfer process of 2D materials. Furthermore, some air-sensitive 2D materials in the ambient atmosphere cause possible oxidation, which leads to the structure rottenness. Additionally, it is very difficult to transfer the 2D materials synthesized on the solid substrates [156].

### 5.2. Transfer Process 

The chemical vapor deposition technique is tremendously used to grow high surface area and the controlled layer of 2D materials on donor substrate, which is a promising functional material for conductors, semiconductors, and insulators in the photodetector, RRAM, gas sensors, flexible, and transparent devices [35]. In several cases, 2D materials must be transferred from a donor substrate to a target substrate; however, there are high possibilities of damaging 2D materials during the transfer process and as a consequence of chemical doping, contamination, and tearing. For the high-performance commercial application, 2D materials should have a high surface area, damage-free structure, and high mechanical and chemical stability. However, a transfer process to overcome all challenges and has still not been developed. Despite this, the transfer process is classified into four parts: wet transfer, dry transfer, mechanical transfer, and electro-chemical transfer [157]. However, the wet transfer process is the most common and widely used process to transfer 2D materials [158]. This process uses polymethyl methacrylate (PMMA) as a transfer film. PMMA plays an extremely significant role in protecting the material from physical damage. It was reported that large surface area 2D material with 4-inch wafer size could be transferred onto desired target substrate using the wet transfer process [159]. After the transfer process, PMMA is difficult to completely remove from the target substrate, which can cause degenerateness of electrical and optical properties of the photodetector fabrication with 2D material. To overcome this challenge and completely remove the PMMA from the material and improve the electrical and optical properties, additional etching and high-temperature annealing have been suggested [160].

### 5.3. The Schottky Barrier

One of the biggest issues confronting 2D materials-based devices is the presence of a Schottky barrier (SB) at the interface between the material and the contact metal electrode. However, the potential energy barrier for electrons formed at the Schottky junction, known as the Schottky barrier height, blocks the flow of charge carriers into the device channel [161]. Subsequently, this prominent Schottky barrier height led to large contact resistance and performance degradation in two-terminal devices [162]. It has been suggested to tune the Schottky barrier by reducing fermi level pinning leads for better performance of devices. Zhou et al. reported the use of graphene as the metal electrode, which comes into contact with single layer of the SnSe sheet to form a van der Waals (vdW) heterojunction. After theoretical calculations, they found that the fermi level pinning is reduced because of the interaction between the SnSe layer and graphene and intrinsic properties of SnSe sheet are protected. Usually, charge carriers move from the source to the channel zone, then they conflict with two energy barriers, one of being the Schottky barrier (*Φ*_SB__⊥_), which can be found at the vertical interface of the contact and used to transport the charge carrier across the interface of the SnSe/G heterojunction, and the other one arises at the lateral interface contact and the channel part characterized by band bending, Δ*E*_F_. These parameters are extremely prominent and inconclusive to the performance of transistors [163].

## 6. Conclusions and Future Outlook

Two-dimensional QDs are increasingly being touted as promising materials due to their fascinating properties (such as full color PL spectrum, high solubility, bandgap tunability, high selectivity to special molecules, and ease of surface functionalization) and widespread applications in sensors, batteries, WLEDs, photodetectors, phototransistors, etc. Despite their great potential for optoelectronics applications, as described in this review article, they encounter a series of daunting obstacles that must be overcome before their commercialization. The emergence of double element QDs, as well as their heteroatom-doped variants, has yet to be thoroughly investigated, both in terms of synthesis and fundamental property evaluation. Several mechanisms in terms of PL improvement, surface passivation, structure optimization, manufacturing aspects, and so on have yet to be established. To fulfill the standards for industrial-scale applications or commercialization, the current output, performance, quantity, and productivity of 2D-QDs must be monitored. Furthermore, energy conversion and storage based on 2D-QDs materials are still in the early stages of development. Because of their relatively low physical and chemical stability, most 2D-QDs currently suffer from long-term stability and durability problems, which must be addressed in the future for practical applications. When 2D-QDs are used as fluorescent active materials, their low quantum yield and wide emission band can pose a problem when compared to conventional fluorophores.

## Figures and Tables

**Figure 1 nanomaterials-11-01549-f001:**
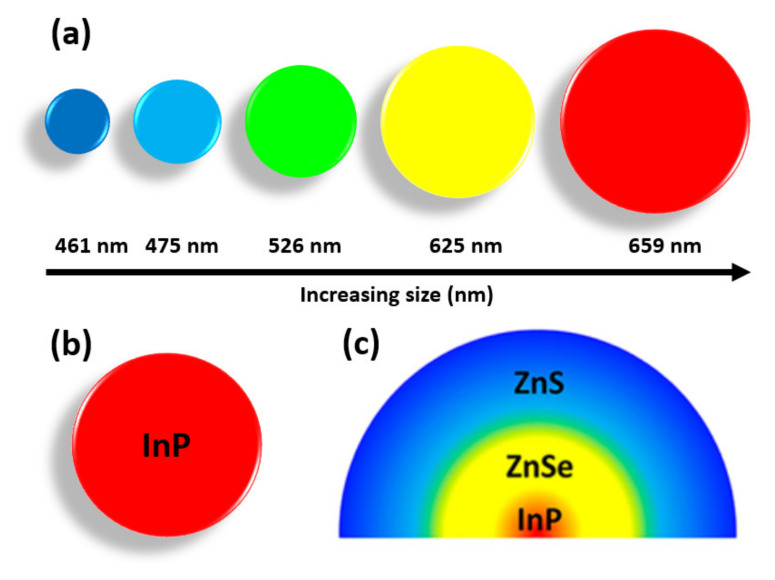
(**a**) Emission color and wavelength of QDs corresponding to their sizes (**b**) InP QDs; (**c**) InP/ZnSe/ZnS core-shell QDs [7]. Figure reproduced with permission from ACS Publications.

**Figure 2 nanomaterials-11-01549-f002:**
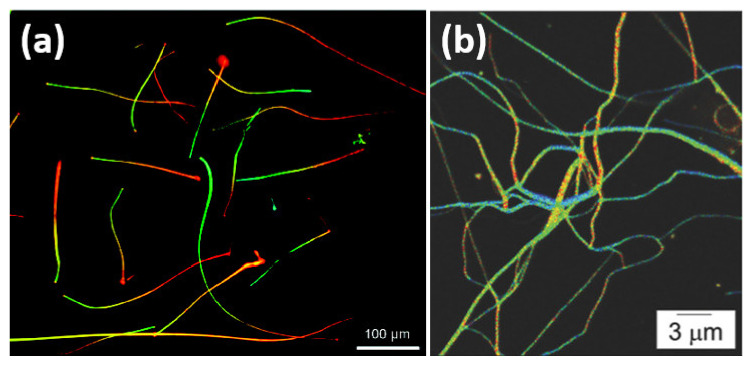
(**a**) CdSSe nanowires dispersed on a low-index MgF_2_ substrate [12]. (**b**) CdSe-CdS quantum rods [13]. Figure reproduced with permission from ACS Publications and Wiley Online Library.

**Figure 3 nanomaterials-11-01549-f003:**
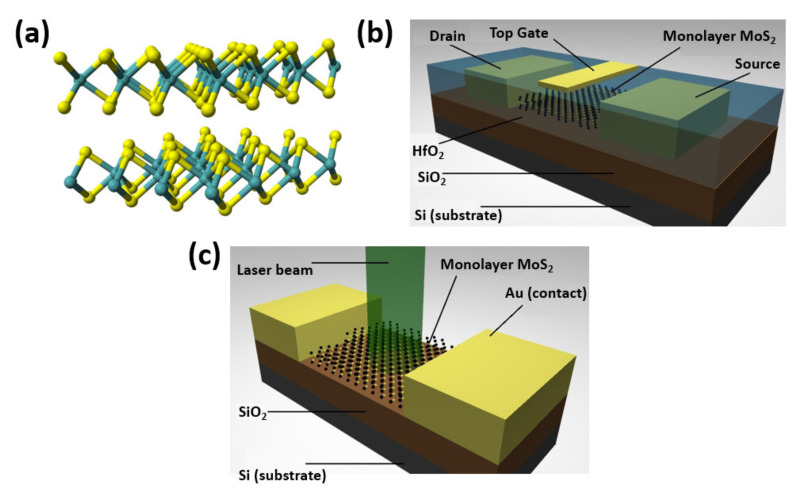
(**a**) Structure of 2-D Quantum dots. (**b**) Field effect transistor based on monolayer of MoS_2_. (**c**) Photodetector based on a monolayer of MoS_2_ [40].

**Figure 4 nanomaterials-11-01549-f004:**
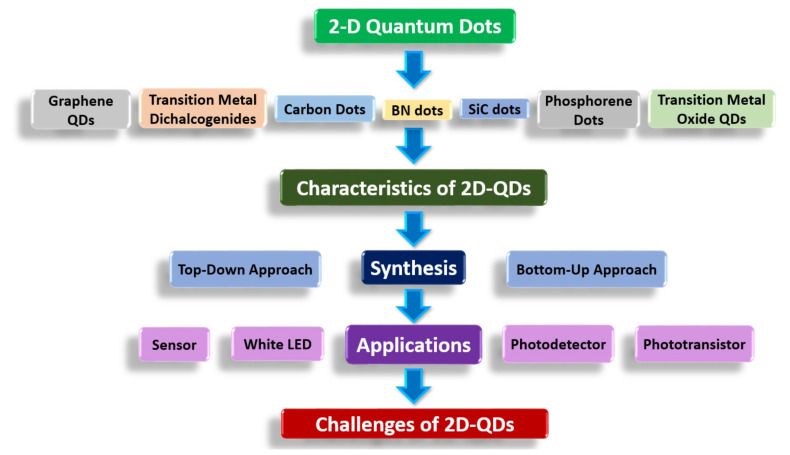
Overview of the present 2-D quantum dots review.

**Figure 5 nanomaterials-11-01549-f005:**
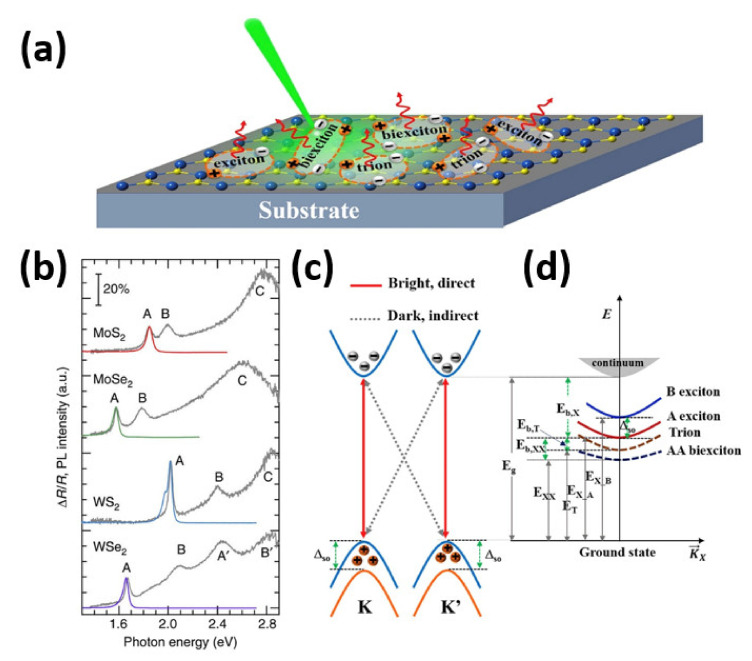
(**a**) Schematic of exciton, trion, biexciton generation in TMDs by photoexcitation. (**b**) PL and differential reflectance spectra of monolayer TMDs flakes on quartz substrate. (**c**) The four possible exciton formations in K and K′ valleys. (**d**) The bound electron-hole pair in exciton picture. [62] Figure reproduced with permission from Elsevier.

**Figure 6 nanomaterials-11-01549-f006:**
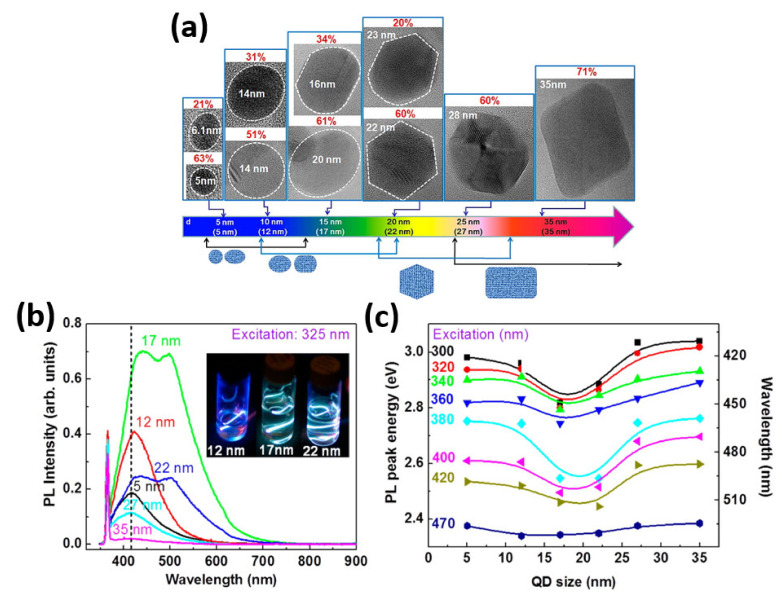
(**a**) HRTEM images of GQDs for their major shapes and corresponding populations (p) with increasing average size of GQDs. (**b**) Size-dependent PL spectra excited at 325 nm for GQDs of 5–35 nm average sizes in DI water. (**c**) Dependence of PL peak shifts on the excitation wavelength from 300 to 470 nm for GQDs of 5–35 nm average sizes. [76] Figure reproduced with permission from ACS Publications.

**Figure 7 nanomaterials-11-01549-f007:**
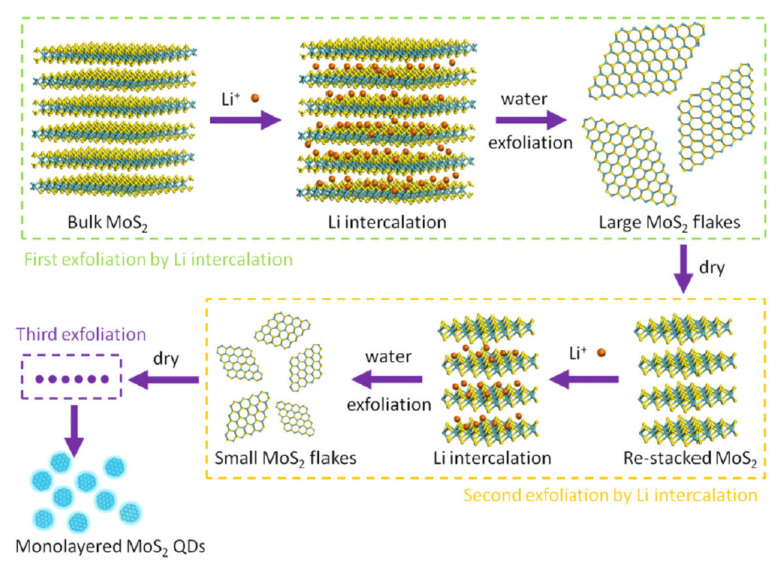
Schematic illustration of the preparation of monolayer MoS_2_ QDs using multiple exfoliation with Li intercalation [78]. Figure reproduced with permission from Elsevier.

**Figure 8 nanomaterials-11-01549-f008:**
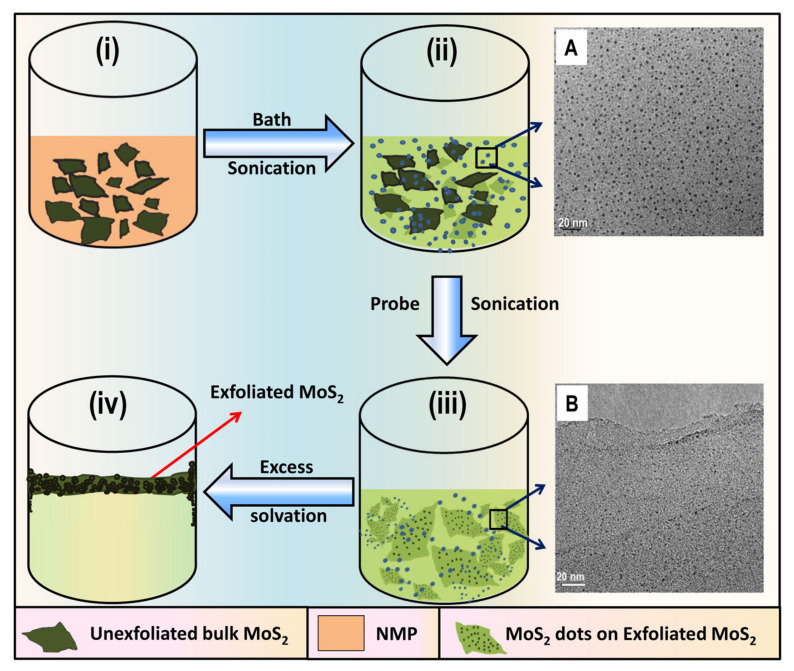
Schematic representation of the synthesis procedure to obtain MoS_2_ quantum dots interspersed in MoS_2_ nanosheets using a liquid exfoliation approach. (**A**) MoS_2_ QDs formed through sonication bath. (**B**) MoS_2_ QDs interspersed in the exfoliated MoS_2_ nanosheets. [79] Figure reproduced with permission from ACS Publications.

**Figure 9 nanomaterials-11-01549-f009:**
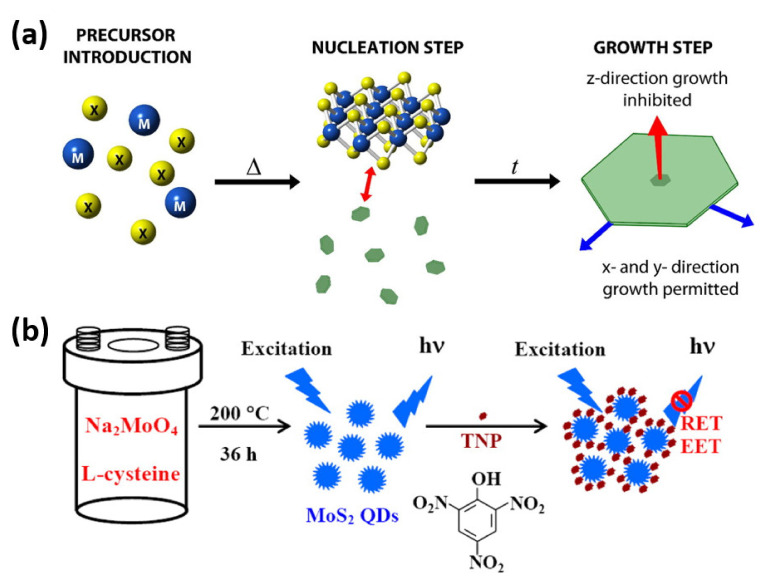
(**a**) Schematic representation of the growth of a nanosheet from molecular precursors [81]. (**b**) Hydrothermal route for the synthesis of photoluminescent MoS_2_ quantum dots (QDs) by using sodium molybdate and cysteine as precursors [82]. Figure reproduced with permission from Elsevier and ACS Publications.

**Figure 10 nanomaterials-11-01549-f010:**
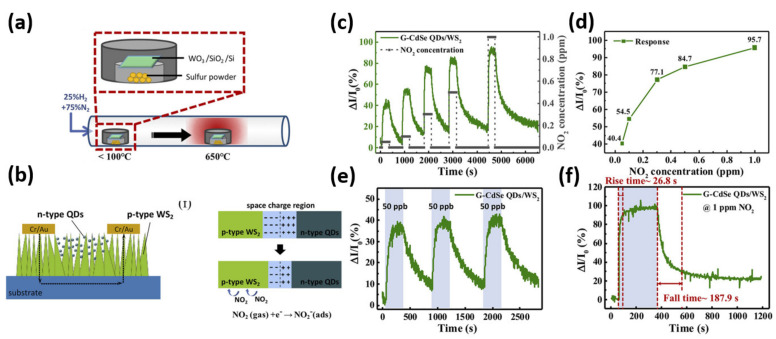
(**a**) Schematic of the synthesis process of WS_2_ nanowalls by the horizontal furnace. (**b**) Structural model and p–n junction model of the QDs/WS_2_ device before and after exposure to NO_2_ gas. (**c**) Time-resolved response measurement of NO_2_. (**d**) Response of the G-CdSe QDs-WS_2_ gas sensor as the function of NO_2_ concentrations. (**e**) Stability test of the NO_2_-gas-sensing properties. (**f**) Rise time and fall time fitting of the G-CdSe QDs-WS_2_ gas sensor. [57] Figure reproduced with permission from ACS Publications.

**Figure 11 nanomaterials-11-01549-f011:**
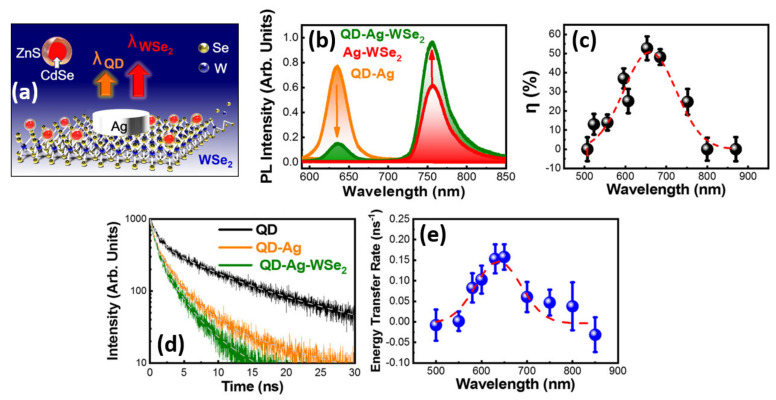
(**a**) Hybrid zero-dimensional core–shell CdSe/ZnS quantum dot (QD)/two-dimensional monolayer WSe_2_ semiconductors with an Ag nanodisk (ND). (**b**) PL spectra of the three samples. (**c**) Energy conversion efficiency from the CdSe QD to WSe_2_. (**d**) PL decay curves of the QD peak (630 nm) in the QD, QD−Ag, and QD−Ag−WSe_2_ structures with an Ag ND. (**e**) Energy transfer rate of QD in the QD−Ag−WSe_2_ structure [58]. Figure reproduced with permission from ACS Publications.

**Figure 12 nanomaterials-11-01549-f012:**
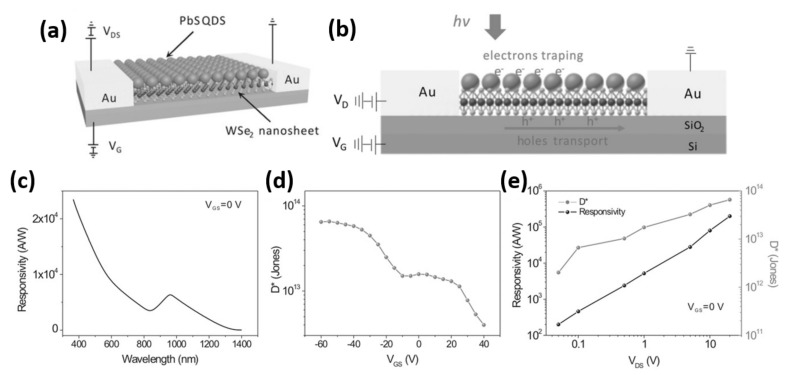
(**a**) Schematic diagram of back-gate WSe_2_ phototransistor device capped by PbS QDs. (**b**) Schematic description for the photocarrier transport in hybrid phototransistor under illumination. (**c**) Wavelength dependence response curve of the hybrid phototransistors with V_DS_ = 1 V. (**d**) Specific detectivity as a function of gate voltage at V_DS_ = 1 V. (**e**) Responsivity and detectivity as a function of V_DS_ (V_GS_ = 0 V). [56] Figure reproduced with permission from John Wiley and Sons.

**Figure 13 nanomaterials-11-01549-f013:**
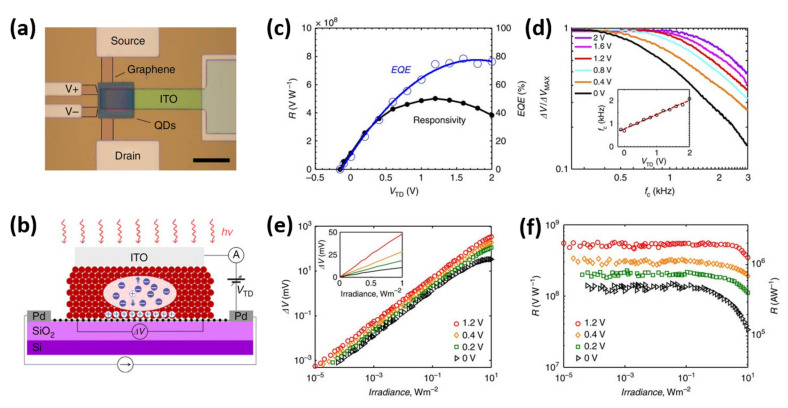
(**a**) Optical image of the hybrid graphene transistor-CQD photodiode detector. (**b**) Schematic of phototransistor operation. (**c**) Responsivity and EQE of the visible/near-infrared phototransistor. (**d**) Normalized photoresponse as a function of light modulation frequency. Inset: extracted 3 dB bandwidth values. (**e**) Photo-induced signal as a function of incident irradiance, inset demonstrates the linearity of photoresponse for high irradiance values. (**f**) Measured responsivity of the detector in VW^−1^ (left axis) and responsivity converted in AW^−1^ (right axis). [145] Figure reproduced with permission from Springer Nature.

**Figure 14 nanomaterials-11-01549-f014:**
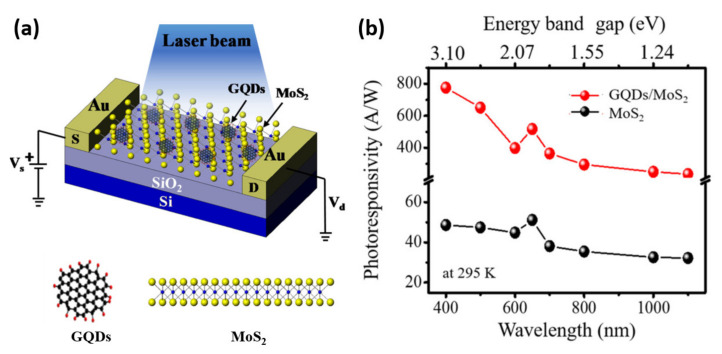
(**a**) Schematic of the hybrid GQD/MoS_2_ device under optical illumination. (**b**) Photoresponsivity of hybrid GQD/MoS_2_ and the bare MoS_2_ devices. [146] Figure reproduced with permission from ACS Publications.

**Table 1 nanomaterials-11-01549-t001:** Performance of 2D-QDs-based devices.

Device Structure	Performance	References
Core-shell QDs/WS_2_ hybrid device for gas sensing	*The hybrid device demonstrates an outstanding NO_2_ gas-sensing performance with a remarkably quick response time of 26.8 s to achieve the outstanding gas response efficiency of 95.7%.*	[57]
CdSe/ZnS QD-Ag-WSe_2_	*The Ag ND was combined with CdSe QDs and monolayer WSe_2_ to enhance the monolayer WSe_2_ emission and convert the light from QDs to WSe_2_ with the highest efficiency of 53%.*	[58]
Hybrid PbS QDs/WSe_2_	*The hybrid device demonstrated a high responsivity up to 2 × 10^5^ A W^–1^ and a high specific detectivity of 7 × 10^13^ Jones.*	[56]
PbS QDs photodiode atop a high-gain graphene phototransistor	*The device demonstrated quantum efficiencies in excess of 70%, gain of 10^5^, and 3 dB bandwidth of 1.5 kHz with a measured detectivity of 1 × 10^13^ Jones.*	[145]
GQD/MoS_2_ hybrid photodetector	*The device exhibits a photoresponsivity 775 AW^–1^ at a laser wavelength of 400 nm, a detectivity of 2.33 × 10^12^ Jones, and an EQE of about 241%.*	[146]
WS_2_ QDs for spintronics and valleytronics	*Ultrasmall and monolayered tungsten dichalcogenide QDs with giant spin–valley coupling and purple luminescence for various applications in spintronics and valleytronics.*	[154]

## Data Availability

The study did not report any data.

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
