# Peer review of "Recent Advances in Two-Dimensional Quantum Dots and Their Applications"

_nanomaterials, 2021, doi:10.3390/nano11061549_

Round 1

Reviewer 1 Report

Authors reported a very comprehensive overview on the cutting edge research represented by  2-D quantum dots. They extensively reported synthetic approaches together a weel described theoretical analysis.  Furthermore, the relevance of deposition in the production of new devices was also considered. 

I suggest to remove the acronyms from the abstract and to improve the quality of figure 3 by magnify teh size of the text there reported.

Anyhow, this is a very noticeble work and i recommed it for pubblication.

Author Response

We thank the reviewer for such beautiful and kind comments. Thanks again for taking the time to review this paper and give us good comments. Regarding the suggestion, we have removed acronyms from the abstract and changed Figure 3 on page 4 with a better resolution. We look forward to writing more effective papers in the future. 

Reviewer 2 Report

While the topic is important, the authors provided only limited contents in this review. 

- The title is broad. If so, the authors should cover broader range of the related topic. 

- Define the "recent", used in the title, in Introduction. 

- The authors are encouraged to define what 2D QDs mean. The 2D indicates the 2D morphology (shape) of the nanocrystals? or the 2D crystallinity?

- The authors are encouraged to make comparison between the 2D and the others (3D?).

-  Regarding the applications, only a few examples are given in this review and, moreover, related explanations are too specific. 

- It would be much better if the authors can address the pros and cons of using 2D QDs in applications. 

Author Response

Point 1. Define the "recent", used in the title, in Introduction. 

Response- We thank the reviewer for the suggestion and we have added a section on page 6 explaining the recent developments for 2D-QDs which will be further explained in detail.

- The authors are encouraged to define what 2D QDs mean. The 2D indicates the 2D morphology (shape) of the nanocrystals? or the 2D crystallinity?

Response- We thank the reviewer for the suggestion and we have defined the 2D QDs by adding a section in the last paragraph of page 3.

- The authors are encouraged to make comparison between the 2D and the others (3D?).

Response- We thank the reviewer for the suggestion and we have added a paragraph for the comparison between 2D and bulk (3D) in the first paragraph of page 3.

-  Regarding the applications, only a few examples are given in this review and, moreover, related explanations are too specific. 

Response- In order to cover some broad area of research for 2D QDs, we have added two sections on pages 20 and 21 discussing GQD/MoS2 photodetector and spin-valley coupling in 2D QDs.

- It would be much better if the authors can address the pros and cons of using 2D QDs in applications. 

Response- We thank the reviewer for the suggestion and we have added a section to discuss the pros and cons of using 2D materials on pages 22 and 23. We already have a section about the challenges faced by the 2D QDs during implementation for various applications.

Overall, we thank the reviewer for the comments and for giving such suggestions to make this paper a better one and we look forward to writing more effective papers in the future.

Reviewer 3 Report

As I got a message there were enough reports fpr the revision process, I will just write a few commenst. The paper is very interesting as a review. I detected a few mistakes (one sentence with no verb page 1 - Introduction- Line 13-14).

I think also the paper could be more readible if the data were shown on tables summarizing each categories. Example of such paper in another field:

Chemosphere, vol. 206, pp. 255-264 (2018).

Author Response

We thank the reviewer for the comments and suggestions. As per reviewer’s suggestions, we have added a table summarizing the performance for each device configuration used in this paper on page 22.

Reviewer 4 Report

The manuscript entitled "Recent Advances in Two-dimensional Quantum Dots
and Their Applications" by Konthoujam James Singh, Tanveer Ahmed,
Prakalp Gautam, Annada Sankar Sadhu, Der-Hsien Lien, Shih-Chen Chen,
Yu-Lun Chueh and Hao-Chung Kuo, about the the characteristics of 2D
QDs materials, their synthesis methods, opportunities and challenges
for novel device applications, is very well done and showing interestind results.
Writting,pictures and figures have a very good quality. Moreover, thew calculations and similutaions are very
well discussed.
Authors concluded that 2D QDs are increasingly being touted as promising
materials due to their fascinating properties (such as full color PL
spectrum, high solubility, bandgap tunability, high selec-tivity to
special molecules, and ease of surface functionalization) and widespread
applications.Because of their relatively low physical and chemical
stability, most 2D QDs currently suffer from long-term stability and
durability problems, which must be addressed in the future for practical
applications.

Author Response

We thank the reviewer for such beautiful and kind comments. Thanks again for taking the time to review this paper and give us good comments. However, we are a bit confused about the remarks given for the 5 questions with 1 star each, and we are unable to get any suggestion from the reviewer’s side. We are highlighting the responses here.

Is the work a significant contribution to the field?    - 1 star

Is the work well organized and comprehensively described? - 1 star

Is the work scientifically sound and not misleading? - 1 star

Are there appropriate and adequate references to related and previous work? – 1 star           

Is the English used correct and readable? – 1 star

If the reviewer is not satisfied with the paper, please let us know some suggestions so that we can make our paper a better one than before. Thanks.

Round 2

Reviewer 2 Report

Now I recommend the manuscript for publication in Nanomaterials.